**Data Availability Statement:** We provided a minimal dataset in the supplementary materials,

# Estimating infection prevalence using the positive predictive value of self-administered rapid antigen diagnostic tests: An exploration of SARS-CoV-2 surveillance data in the Netherlands from May 2021 to April 2022

**Koen M.F. Gorgels**[1,2,3]*, **Senna C.J.L. van Iersel**[1], **Sylvia F.A. Keijser**[1], **Christian J.P.A. Hoebe**[2,3,4], **Jacco Wallinga**[1], **Albert J. van Hoek**[1]

1 Epidemiology and Surveillance Unit, Centre for Infectious Diseases Control, National Institute for Public Health and the Environment (RIVM), Bilthoven, The Netherlands, 2 Department of Sexual Health, Infectious Diseases and Environmental Health, Living Lab Public Health, South Limburg Public Health Service, Heerlen, The Netherlands, 3 Department of Social Medicine, Care and Public Health Research Institute (CAPHRI), Maastricht University, Maastricht, The Netherlands, 4 Department of Medical Microbiology, Infectious Diseases and Infection Prevention, Care and Public Health Research Institute (CAPHRI), Maastricht University Medical Centre (MUMC+), Maastricht, The Netherlands

* Koengorgels@gmail.com

## Abstract

Measuring the severity of the disease of SARS-CoV-2 is complicated by the lack of valid estimations for the prevalence of infection. Self-administered rapid antigen diagnostic tests (Ag-RDTs) were available in the Netherlands since March 2021, requiring confirmation by reverse-transcription polymerase chain reaction (RT-PCR) for positive results. We explored the possibility of utilizing the positive predictive value (PPV) of Ag-RDTs to estimate SARS-CoV-2 prevalence. We used data from all Public Health service testing facilities between 3 May 2021 and 10 April 2022. The PPV was calculated by dividing the number of positive RT-PCR results by the total number of confirmation tests performed, and used to estimate the prevalence and compared with the number of COVID-19 hospital admissions. In total 3,599,894 cases were included. The overall PPV was 91.8% and 88.8% were symptomatic. During our study period, the estimated prevalence ranged between 2–22% in symptomatic individuals and 2–14% in asymptomatic individuals, with a correlation between the estimated prevalence and hospital admissions two weeks later (r = 0.68 (p<0.01) and r = 0.60 (p<0.01) for symptomatic/asymptomatic individuals). The PPV of Ag-RDTs can help estimate changes in SARS-CoV-2 prevalence, especially when used in conjunction with other surveillance systems. However, the used method probably overestimated the true prevalence because of unmonitored differences in test propensity between individuals.

with information for each specific week on symptom status and age group. All analyses and visual representations in our study were derived from this fully anonymized dataset.

**Funding:** The study was financed by the Netherlands Ministry of Health, Welfare and Sport. The funders had no role in study design, data collection and analysis, decision to publish, or preparation of the manuscript.

**Competing interests:** The authors have declared that no competing interests exist.

## Introduction

In a pandemic scenario, it is imperative to closely monitor the prevalence of SARS-CoV-2 infections, or any potential future pandemic pathogen. While hospital admissions can serve as an indicator of prevalence, accurate estimations of the underlying prevalence of infection or the infection hospitalization ratio remain elusive. Moreover, the growing disparities within the population in terms of prior infection and vaccination, along with their potential influence on disease severity and symptoms, adds another layer of complexity to these estimations. In this paper, we investigate the potential utility of employing the positive predictive value (PPV) of self-administered rapid antigen diagnostic tests (Ag-RDTs) to gauge the underlying prevalence and inform the debate on disease severity.

While reverse-transcription polymerase chain reaction (RT-PCR) is considered the reference standard for diagnosing SARS-CoV-2 infection, Ag-RDTs are cheap, fast and simple to use, making it possible for a lot of individuals to test themselves without the need to wait for the results of RT-PCR [1, 2]. Over the counter Ag-RDTs have been available for sale in the Netherlands since March 2021 and were recommended as a preventive screening measure for asymptomatic individuals for school, work, or when returning from travel abroad. The Dutch government also distributed self-tests throughout the Netherlands for free use in e.g. the education system, Salvation Army, and Red Cross [3, 4]. Starting 3 December 2021, the recommendation also included self-testing when developing COVID-like symptoms, or prior to social visits. Important for this study: any positive result from an Ag-RDT needed to be confirmed by RT-PCR at the Municipal Public Health Services (PHS) since the test becoming available. From 11 April 2022, when self-testing became the main gauge to determine advice on behaviour, diagnostics, isolation and quarantine for the general population, the confirmation requirement was cancelled [5].

Because the specificity of Ag-RDTs is not 100%, the positive predictive value (PPV, defined as the probability that an individual with a positive Ag-RDT for SARS-CoV-2 also tests positive in the reference method) of Ag-RDTs is not 100%. Even in a situation with no prevalence of SARS-CoV-2 false positives will occur in Ag-RDT testing. As prevalence of infection rises, one would anticipate a decrease in the number of false positives and an increase in the PPV. Therefore, by tracking the PPV, one can infer insights about the prevailing prevalence. Such an estimate of prevalence is highly relevant as it could be used to measure the IFR and IHR and become part of the SARS-CoV-2 surveillance. In this paper the possibility to utilize the PPV of Ag-RDTs to estimate the prevalence of SARS-CoV-2 infection is explored, with Dutch data from May 2021 to April 2022.

## Methods

In this study, we evaluate the use of the PPV of Ag-RDTs by inferring a prevalence and comparing this to hospital admissions, over time, in Dutch surveillance data. Furthermore, we perform a sensitivity analysis deconstructing the prevalence estimation.

### Timeline during our study period

Throughout a substantial portion of our study period, non-pharmaceutical measures against the spread of SARS-CoV-2 were implemented in the Netherland. These measures encompassed mandatory mask-wearing in crowded settings, including public transportation, along with the advice to uphold a 1.5-meter physical distance. The catering industry faced either closures or stringent regulations, which included limitations on customer capacity and controlled access through the implementation of a corona entry certificate. In June 2021, a brief respite from these restrictions was granted, only to be swiftly reinstated two weeks later in response to

a rapid surge in infection rates followed by cautiously easing until November 2021, when non-essential stores and all establishments within the catering industry were temporarily shut. Eventually, in February 2022, most restrictions were lifted, including the discontinuation of the corona entry certificate.

## Ethical statement

In accordance with Dutch law, approval from a medical ethics committee was not deemed necessary since cases were not subject to any actions or rules of conduct. Data regarding cases were obtained by use of standard surveillance tasks of the RIVM (Law on the RIVM, article 3. Article 6c of the Public Health Act (Wet Publieke Gezondheid), and pseudonymized data were used in the study [6]. Informed consent was not obtained, as the collection of data complies with the exceptions for not asking informed consent as formulated in the Dutch Implementation Act General Data Protection Regulation.

## Calculating PPV and prevalence

Data to calculate the PPV was obtained from all PHS testing facilities in the Netherlands over the period 3 May 2021 (2021-W18) until 10 April 2022 (2022-W14). Every individual who underwent testing because of a self-reported positive self-administered Ag-RDT was included. Researchers had no access to information that could identify individual participants. To assure that the interpretation of the PPV was valid we applied the following criteria: the confirmation test had to be an RT-PCR test, indeterminate RT-PCR results and an RT-PCR conducted more than 48 hours after the positive Ag-RDT were excluded. Furthermore, situations where the date of the Ag-RDT was not known were also excluded. The PPV was calculated per week by dividing the number of positive RT-PCR results by the total number of included confirmation tests performed.

The PPV can also be calculated using the prevalence and test characteristics: PPV = (sensitivity * prevalence) / [(sensitivity * prevalence) + ((1 –specificity) x (1 –prevalence)) ] [7]. Using this formula we can derive the prevalence using the PPV: Prevalence = (1- specificity)/ ((sensitivity / PPV) - sensitivity–specificity + 1). The prevalence is 100% if the PPV is 100%. A lower PPV leads to an increasingly lower prevalence but a PPV of zero can't be used.

In estimating the prevalence we differentiated between those who were symptomatic, and those who were asymptomatic. This is because information was available on symptomatic disease, and the sensitivity of Ag-RDT is known to be different for both. Sensitivity was set at 0.8 for symptomatic individuals, 0.5 for asymptomatic individuals and specificity at 0.99 for both, based on multiple Western-European studies on self-administered Ag-RDTs [8–11].

Publicly available data on the number of COVID-19 hospital admissions was plotted alongside the estimated prevalence as a proxy for prevalence [12]. A Pearson coefficient was calculated to examine the overall correlation between the estimated prevalence and hospital admissions. Due to decreased virulence of the SARS-CoV-2 Omicron variant the correlation was evaluated during the period in which Omicron was the dominant SARS-CoV-2 variant in circulation (from 3 January 2022 (2022-W01)). To account for a potential delay between infection and hospital admission the correlation between hospital admissions and four different prevalence values was evaluated, i.e. current estimated prevalence, and prevalence one, two and three weeks earlier. Additional analyses were performed by dividing cases based on symptom status and age (split up in four groups: 0–17, 18–39, 40–59 and 60+) and using the PPV per week to estimate the prevalence.

All data was analyzed in RStudio version 2021.09.1+372 [13]. In our analysis 95% binomial confidence intervals were created using count data. In the descriptive summary we assumed 5

days of unnecessary isolation were prevented per negative confirmation test, based on Dutch guidelines.

## Sensitivity analysis

The estimated PPV using population data is affected by any factor that influences the probability of performing an Ag-RDT or being SARS-CoV-2 positive. Therefore, the effect of various dependent factors on the PPV and prevalence was explored including symptom status, percentage of people with symptoms, prevalence of COVID-like symptoms in SARS-CoV-2 negative individuals and the propensity to test for SARS-CoV-2 positive and SARS-CoV-2 negative individuals. Test propensity was defined as the probability to perform an Ag-RDT and subsequently get a confirmation RT-PCR test at the PHS. Test propensity was included because it is different between SARS-CoV-2 positive and negative individuals due to information from the social network (contact being positive), variation in exposure or severity of symptoms. The equations used in this analysis are found in the S1 File.

The proportion symptomatic (percentage of all infected individuals that develops symptoms) was set at 60% [14]. Using data from syndromic surveillance in the Netherlands (a national weekly online questionnaire among 15.000 to 30.000 participants since March 2020) and data from the RIVM Corona Behavioural Unit (national repeated online questionnaires among 37.000 to 64.000 participants) the percentage of the population with COVID-like symptoms & SARS-CoV-2 negative was set at 3%, propensity to test was set at 50% for symptomatic individuals and 7% for asymptomatic individuals [15, 16]. Sensitivity of Ag-RDT's was set at 0.8 for symptomatic individuals, 0.5 for asymptomatic individuals and specificity at 0.99 for both groups [8–11].

Values were experimentally varied to examine the size and direction of any effect on the true and estimated prevalence of both symptomatic and asymptomatic individuals.

## Results

### Number of confirmation tests in study period

A number of 3,949,332 cases underwent testing after a positive self-administered Ag-RDT. After excluding non RT-PCR tests (0.4%), indeterminate results (<0.1%) and RT-PCR conducted >48 hours after the Ag-RDT or unknown Ag-RDT date (8.5%), 3,599,894 cases remained. Overall positivity rate was 91.8% with 308,017 cases having a negative RT-PCR preventing approximately 1.5 million days of unnecessary isolation (Table 1). The majority (88.8%) was symptomatic, mostly nose cold, coughing and throat pain.

### PPV and estimated prevalence

Over the study period the PPV ranged between 60–96% for symptomatic individuals and between 48–90% in asymptomatic individuals (Fig 1). A difference in PPV ranging between 6–19% can be observed between symptomatic and asymptomatic individuals with the former having a higher PPV regardless of time period.

The estimated prevalence ranged between 2–22% in symptomatic individuals and 2–14% in asymptomatic individuals (Fig 2). Fig 2 shows the estimated prevalence using the PPV and the hospital admissions per week. In contrast to the PPV (Fig 1), some weeks there was no difference in estimated prevalence between symptomatic and asymptomatic individuals. The overall correlation between prevalence and hospital admissions was highest when looking at the estimated prevalence two weeks earlier (r = 0.68 (p<0.01) for asymptomatic individuals and r = 0.60 (p<0.01) for symptomatic individuals). In the period 18 October 2021 till 12

**Table 1. Characteristics study population.**

| | |
|---|---|
| **Total** | **3,599,894** |
| **Result RT-PCR** | |
| Positive | 3,303,486 (91.8%) |
| Negative | 308,017 (8.2%) |
| **Sex** | |
| Male | 1,734,482 (48.2%) |
| Female | 1,863,909 (51.8%) |
| Unknown | 1503 (0.0%) |
| **Age group** | |
| 0–17 | 766,955 (21.3%) |
| 18–39 | 1,459,560 (40,5%) |
| 40–59 | 957,618 (26.6%) |
| 60+ | 414,951 (11.5%) |
| Unknown | 810 (0.0%) |
| **Symptomatic** | 3,195,409 (88.8%) |
| • Nose cold | 2,236,539 (70.0%) |
| • Sore throat | 1,896,309 (59.3%) |
| • Coughing | 1,845,731 (57.8%) |
| • Fever | 930,102 (29.1%) |
| • Shortness of breath | 439,061 (13.7%) |
| • Loss of taste | 204,968 (6.4%) |
| • Loss of smell | 197,688 (6.2%) |
| • Myalgia | 28,738 (0.9%) |
| **Asymptomatic** | 404,485 (11.2%) |
| **Vaccination status** | |
| Booster* | 957,443 (26.6%) |
| Fully vaccinated** | 888,199 (24.7%) |
| Partly vaccinated*** | 153,579 (4.3%) |
| Unvaccinated | 1,057,391 (29.4%) |
| Unknown vaccination status | 543,282 (15.1%) |

\* = >1 weeks after receiving a booster vaccination

\*\* = >2 weeks after receiving final vaccination or > = 4 weeks after receiving Janssen vaccination.

\*\*\* = All individuals with at least one vaccination that do not fulfill the criteria of the fully vaccinated category.

December 2021 (2021-W42 till 2021-W49 the estimated prevalence dropped while hospital admissions rose from 100 to a peak of 300. Afterwards the estimated prevalence rose again and from 3 January 2022 (2022-W01) onwards (start of omicron period) the correlation peaked between prevalence and hospital admissions one week later ($r = 0.84$ ($p<0.01$) for asymptomatic individuals and $r = 0.74$ ($p<0.01$) for symptomatic individuals). An overview of the correlations and an additional analysis showing similar patterns of estimated prevalence occurring in all age groups are included in the S1 Fig and S1 Table.

## Sensitivity analysis

Our sensitivity analysis shows that an increase in overall test propensity did not change the estimated prevalence if the test propensity was equal between SARS-CoV-2 positive and negative individuals but an increase in test propensity for SARS-CoV-2 positive individuals relative to the test propensity for SARS-CoV-2 negative individuals resulted in an overestimation of

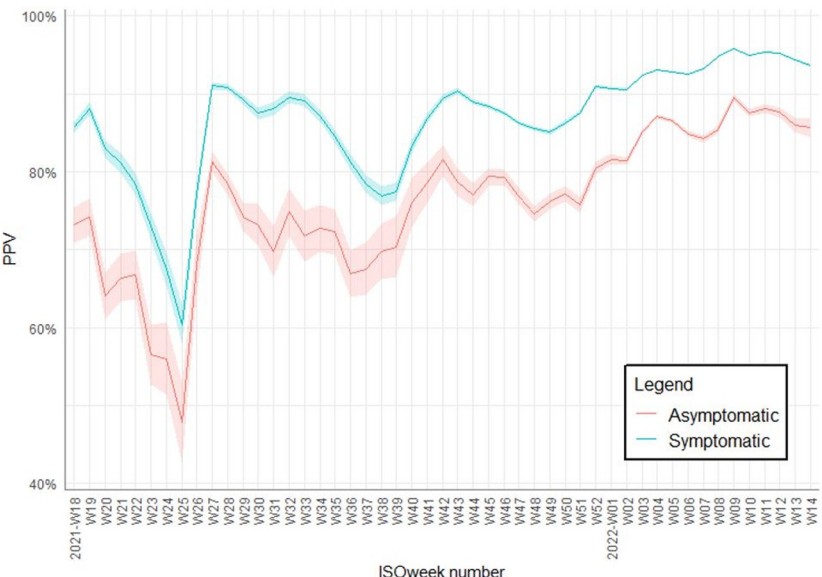

**Fig 1. Overall positive predictive value (PPV) of a positive Ag-RDT split up by COVID-like symptomatic status with 95% confidence intervals included in the shaded lines.**

the prevalence (Table 2). An increase in the prevalence of COVID-like symptoms (without an increase in SARS-CoV-2 prevalence) leads to a lower true and estimated prevalence in symptomatic individuals. An underestimation of the sensitivity and specificity of the Ag-RDT's leads to an overestimation of the prevalence. Analysis conducted on data from the RIVM Corona Behavioural Unit shows indeed an increased propensity to test after known exposure to SARS-CoV-2 (S2 Fig).

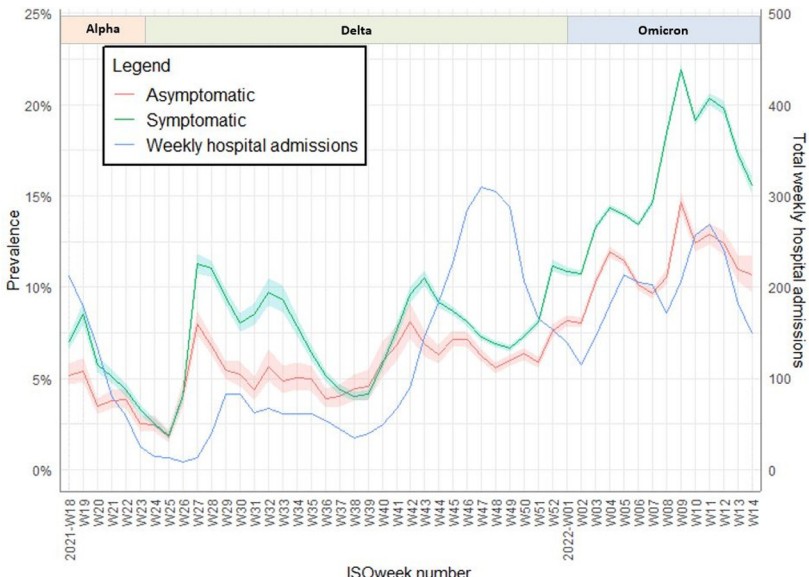

**Fig 2. Prevalence estimated using the positive predictive value (PPV) per COVID-like symptomatic status and including COVID-related hospital admissions per week.** A sensitivity of 0.8 was used in symptomatic individuals and 0.5 in asymptomatic individuals. The dominant variants are displayed per week.

**Table 2. Results sensitivity analysis on how changing baseline variables regarding SARS-CoV-2 and COVID-like symptoms influence modelled prevalence results.**

| | Baseline variables | | | | | Results | | | |
|---|---|---|---|---|---|---|---|---|---|
| | Overall Prevalence | Percentage Symptomatic | % population COVID-like symptoms but SARS-CoV-2 - | Propensity to test SARS-CoV-2 positive / SARS-CoV-2 negative (symptomatic) | Propensity to test SARS-CoV-2 positive / SARS-CoV-2 negative (asymptomatic) | True prevalence symptomatic | Estimated prevalence symptomatic | True prevalence asymptomatic | Estimated prevalence symptomatic |
| Baseline | 1% | 60 | 3 | 50%/50% | 7%/7% | 16.8% | 16.8% | 0.4% | 0.4% |
| **Change from baseline** | | | | | | | | | |
| Increase in prevalence | 2%* | 60 | 3 | 50%/50% | 7%/7% | 29.0% ↑ | 29.0% ↑ | 0.8% ↑ | 0.8% ↑ |
| Increase in population COVID-like symptoms | 1% | 60 | 6* | 50%/50% | 7%/7% | 9.7% ↓ | 9.7% ↓ | 0.4% | 0.4% |
| increased overall propensity to test | 1% | 60 | 3 | 80%*/80%* | 15%*/15%* | 16.8% | 16.8% | 0.4% | 0.4% |
| Increased test propensity SARS-CoV-2 positive | 1% | 60 | 3 | 80%*/50% | 15%*/7% | 16.8% | 24.4% ↑ | 0.4% | 0.9% ↑ |
| True sensitivity is 0.9 in symptomatics and 0.6 in asymptomatics** | 1% | 60 | 3 | 50%/50% | 7%/7% | 16.8% | 18.5% ↑ | 0.4% | 0.5% ↑ |
| True specificity is 0.995* | 1% | 60 | 3 | 50%/50% | 7%/7% | 16.8% | 33.6% ↑ | 0.4% | 0.8% ↑ |

* Increase from baseline scenario.

**Baseline scenarios were calculated assuming a sensitivity of 0.8 in symptomatic individuals, 0.5 in asymptomatic individuals and a specificity of 0.99

## Discussion

The results of our study show that the PPV of self-administered Ag-RDTs can be used to estimate changes in the prevalence of the number of SARS-CoV-2 infections. A significant correlation was determined between the estimated prevalence and hospital admissions two weeks later in both symptomatic and asymptomatic individuals. However, the used method probably overestimated the true prevalence due to various factors including increased test propensity in SARS-CoV-2 positive individuals and therefore has limited use to infer disease severity.

The PPV of a self-administered Ag-RDT fluctuated over time and is generally above 75% in symptomatic individuals with an average PPV of almost 92%. In asymptomatic individuals the PPV was consistently lower but not below 48%, even in times of low prevalence. Based on this study we conclude that the PPV can be used for an estimation of the prevalence. However, it remains important to do this alongside other surveillance tools, which monitor relevant behaviour, syndromic surveillance, waste water monitoring and sentinel general practitioner practices to monitor SARS-CoV-2 prevalence [17]. To optimize the use of the PPV of Ag-RDTs we advise that the use of self-tests can be protocolized, for example in the context of a large cohort study in which also the context of use (exposed/non-exposed) can be monitored and the brand/type of Ag-RDTs is standardized. Performing a RT-PCR confirmation test after a positive Ag-RDT has benefits for the individual and society because in the case of a false positive test unnecessary isolation is prevented and this reduces social isolation and improves (labour)

productivity. We estimated that by detecting false positive Ag-RDT's by RT-PCR confirmation the societal costs of approximately 1.5 million days of unnecessary isolation were prevented. Whether this outweighs the financial costs of widespread RT-PCR tests is beyond the scope of this study but our data can be used in subsequent policy evaluations. This study also provides a basis for future research with the PPV of Ag-RDTs in various infectious diseases as we have shown various factors that affect the PPV.

Being the only research to our knowledge that looked at the utility of estimating the prevalence based on the PPV of Ag-RDTs in the general population, this study is based on a large dataset collected as part of the national pandemic response. The major limitation of this study is that we calculated the PPV and the estimated prevalence among those who came forward for testing. Populations who did not test were not included in our analysis. Furthermore, the interpretation of the prevalence among asymptomatic and symptomatic individuals is not straightforward because the sampled population and the test propensity is not clear. One could interpret those asymptomatic as everyone without having acute respiratory symptoms. In the syndromic surveillance performed at our institute it is suggested that this applies to over 90% of the population at any given point during the pandemic [15]. Hence, when we observe a roughly similar prevalence in the symptomatic and asymptomatic it means that the vast majority of infections (in absolute numbers) should happen among the asymptomatic, in a ratio of at least 9 to 1. Other estimations of the percentage of asymptomatic infections are around 40% so we believe our suggested ratio between asymptomatic and symptomatic infection is not true [14].

Furthermore, among those with symptoms certain symptoms might have triggered testing in some, but not others, including symptoms such a lack of taste and smell, which are more uniquely linked to a SARS-CoV-2 infection compared to more general symptoms such as cough or runny nose which leads to a relatively higher prevalence compared to a situation where testing is more rigidly linked to a specific set of symptoms. Although Ag-RDTs were advertised to use in situations without symptoms, for example before visiting family or attending a concert, attending such events also increases the chance of an infection and hence we doubt it that people who actually used Ag-RDTs can be seen as a random sample from the population. Furthermore, vaccination or previous SARS-CoV-2 infection influenced the test propensity. Exposure to an individual with SARS-CoV-2 increased the propensity to test and also increased the chance of SARS-CoV-2 infection, as observed in a secondary analysis of a Corona behavioral survey at our institute. In our study no data was available on exposure. In line, our sensitivity analysis shows that a higher test propensity in SARS-CoV-2 positive individuals relative to SARS-CoV-2 negative individuals leads to an overestimation of the prevalence.

We believe that our estimations of the prevalence overestimate the true prevalence, especially our estimate in asymptomatic individuals and therefore this estimate should not be used when better estimates are available. The estimate for this group ranged between 2–14% and overlapped with the estimated prevalence in symptomatic individuals at multiple points during our study. In contrast, earlier estimations of the overall prevalence in the Netherlands ranged between 0–1% of the population and research from other countries suggest point prevalence never topped 2–3% [18–20]. The relation between PPV and prevalence is not linear, but such that for a low PPV the prevalence changes little, but for higher values of the PPV the prevalence changes very rapidly (S3 Fig). Therefore small increases of a higher PPV (due to for example a higher test propensity among individuals infected with SARS-CoV-2) can lead to steep increases in the estimated prevalence. Besides test propensity, our estimates of test characteristics could also have impacted the validity of the estimated prevalence.

Nevertheless, a significant correlation was found between the estimated prevalence and number of hospital admissions suggesting that the PPV in the study population can be used to

estimate changes of the prevalence in the entire population. An earlier study supports the relationship between PPV and prevalence [21]. The number of hospital admissions is not a perfect proxy for prevalence because it is an incidence rate. Besides infection prevalence other factors impact this rate including age, immunity due to vaccination or previous infections and changing virulence across variants. Overall a similar trend in observed prevalence was observed in all age groups. Due to decreased virulence of the Omicron variant for example, a lower number of hospital admissions is expected when prevalence remains equal which matches our results. It appears that a change in estimated prevalence is followed by a change in hospital admissions approximately two weeks later. This time period fits the modelled delay between infection and hospital admission [22]. For the omicron period this delay appears to drop to one week, as correlation was highest when comparing estimated prevalence and hospital admissions one week later, and can be explained by a faster incubation rate of the Omicron variant [23, 24].

Additional factors, including data on relevant behaviour, syndromic surveillance data and policy changes, need to be considered if the PPV is used to estimate trends in underlying prevalence. A drop in PPV and estimated prevalence was seen during a steep rise in hospital admissions from 2021-W42 to 2021-W49. Increased awareness of circulating SARS-CoV-2 may have increased Ag-RDT use preventively, thereby reducing the PPV. A strong rise in the use of Ag-RDTs among all age groups in this period supports this [25]. A (seasonal) increase in COVID-like symptoms due to other pathogens could also have contributed by increasing the test propensity of SARS-CoV-2 negative individuals. The initial advice to only use self-tests when asymptomatic may also have caused people with multiple or severe symptoms to directly visit the PHS and skip the use of Ag-RDTs, while those with less distinctive symptoms (and at a lower chance of SARS-CoV-2 infection) opted for a self-test instead. After 3 December 2021 (2021-W48) an Ag-RDT was also recommended for symptomatic individuals and the PPV rose in the period afterwards supporting this. These factors are less relevant for asymptomatic individuals, potentially accounting for the stronger correlation observed in that group.

This study provides a basis for future research with the PPV of Ag-RDTs in various infectious diseases. Many factors have been shown to influence the PPV. The PPV can be used to examine epidemiological and behavioural differences in subgroups based on age, sex, vaccination status and more. Future research could also further examine the estimated prevalence and see if correcting for known factors is possible to more accurately estimate the true prevalence.

To conclude, this study shows that there is potential for an estimation of the prevalence using the PPV to be included in surveillance tools, especially when used in conjunction with other surveillance systems. Our research also opens up the possibility of using differences in PPV to examine epidemiological and behavioural differences in subgroups such as sex or vaccination status.

## Supporting information

**S1 File. Formula used in our sensitivity analysis.**
(PDF)

**S2 File. Underlying minimal dataset.**
(XLSX)

**S1 Table. Correlation between COVID-related hospital admissions and estimated SARS-CoV-2 prevalence in the population at different points in time.**
(PDF)

**S1 Fig. Estimated prevalence in COVID-like symptomatic individuals per age group, using the positive predictive value.**
(TIF)

**S2 Fig. Test propensity in COVID-like symptomatic individuals, per exposure status, as registered by the RIVM Corona Behavioural Unit.**
(TIF)

**S3 Fig. Mathematical relationship between positive predictive value (PPV) and prevalence for symptomatic and aymptomatic populations.** Sensitivity was set at 0.8 for symptomatic individuals, 0.5 for asymptomatic individuals and specificity at 0.99 for both groups.
(TIF)

## Author Contributions

**Conceptualization:** Koen M.F. Gorgels, Senna C.J.L. van Iersel, Sylvia F.A. Keijser, Albert J. van Hoek.

**Data curation:** Koen M.F. Gorgels, Senna C.J.L. van Iersel.

**Formal analysis:** Koen M.F. Gorgels, Senna C.J.L. van Iersel, Albert J. van Hoek.

**Investigation:** Koen M.F. Gorgels, Sylvia F.A. Keijser, Jacco Wallinga, Albert J. van Hoek.

**Methodology:** Koen M.F. Gorgels, Sylvia F.A. Keijser, Christian J.P.A. Hoebe, Jacco Wallinga, Albert J. van Hoek.

**Project administration:** Christian J.P.A. Hoebe, Albert J. van Hoek.

**Supervision:** Senna C.J.L. van Iersel, Christian J.P.A. Hoebe, Jacco Wallinga.

**Validation:** Christian J.P.A. Hoebe, Albert J. van Hoek.

**Writing – original draft:** Koen M.F. Gorgels, Sylvia F.A. Keijser.

**Writing – review & editing:** Koen M.F. Gorgels, Senna C.J.L. van Iersel, Sylvia F.A. Keijser, Christian J.P.A. Hoebe, Jacco Wallinga, Albert J. van Hoek.

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
