## [Decision Letter · Decision Letter 0]

6 Sep 2023

PONE-D-23-14636Estimating infection prevalence using the positive predictive value of self-administered rapid antigen diagnostic tests: an exploration of SARS-CoV-2 surveillance data in the Netherlands from May 2021 to April 2022PLOS ONE

Dear Dr. Gorgels,

Thank you for submitting your manuscript to PLOS ONE. After careful consideration, we feel that it has merit but does not fully meet PLOS ONE’s publication criteria as it currently stands. Therefore, we invite you to submit a revised version of the manuscript that addresses the points raised during the review process.

ACADEMIC EDITOR:

Kindly consider the comments from myself & another reviewers. 

We look forward to receiving your revised manuscript.

Kind regards,

Liling Chaw

Academic Editor

PLOS ONE

Journal Requirements:

Additional Editor Comments:

I have also acted as a 2nd reviewer, please find below my comments.

Overall, the topic is interesting & definitely of value to the scientific community, particularly in terms of pandemic preparedness. However, I have several points to help improve the manuscript.

Methods

The ethics statement is missing in the main text.

Results

(lines 154-61): I find it quite interesting that the correlation is positively higher for asymptomatic cases than for symptomatic cases. Could the authors explain this finding?

Under line (174-5) “An underestimation of the sensitivity and specificity of the Ag-RDT’s leads to an overestimation of the prevalence” : Is this sentence reporting the table results or is this a discussion point? If the former is true, I’m not sure if this is reporting the last 2 rows of Table 2? Also, I would suggest adding the sensitivity & specificity values for the baseline as a Table footnote.

Discussion

While the relationship between PPV and severity is emphasized in the introduction (and which is likely the basis why COVID-related hospital admissions data were collected), I feel there is a lack in discussing this point in the discussion section. One reason could be the structure of the section. This is especially true for the 6th paragraph, I’m quite confused why hospital admissions are being compared with prevalence. I would suggest to consider revising the discussion section to better clarify the points intended by the authors.

Suggest to include a brief explanation, or a timeline figure, on the public health intervention measures implemented in the Netherlands during the study period. As this would help readers understand better the local context and also the points raised in the discussion section.

Minor comments:

- Abstract (2nd paragraph, 2nd sentence): Please clarify it’s the overall positivity rate of..?

- The manuscript has some minor punctuation mistakes; I suggest to proofread carefully

- Table 1 caption: I don’t understand this, suggest to clarify: “***= All individuals with at least one vaccination that are not covered by the fully vaccinated category.”

- Table 2 (columns 5 & 6): Please check the headings if this should this be correctly read as “Propensity to test SARS-CoV-2 + / SARS-CoV-2 - ?” I would suggest to use positive & negative to avoid confusion, or add a footnote to clarify.

Reviewers' comments:

Reviewer's Responses to Questions

**Comments to the Author**

1. Is the manuscript technically sound, and do the data support the conclusions?

Reviewer #1: Yes

2. Has the statistical analysis been performed appropriately and rigorously? 

Reviewer #1: Yes

3. Have the authors made all data underlying the findings in their manuscript fully available?

Reviewer #1: Yes

4. Is the manuscript presented in an intelligible fashion and written in standard English?

Reviewer #1: Yes

5. Review Comments to the Author

Reviewer #1: Generally, well written manuscript. with the vailability of vaccination data for more than two thirds of study participants it would be great to get a sense of how vaccination coverage over time impacted estimated prevalence/ PPV in the vaccinated compared to the unvaccinated groups

6. PLOS authors have the option to publish the peer review history of their article (what does this mean?). If published, this will include your full peer review and any attached files.

Reviewer #1: No

---

## [Author Response · Author response to Decision Letter 0]

30 Oct 2023

We made these changes

We provided a minimal dataset in the supplementary materials, with information for each specific week on symptom status and age. All analyses and visual representations in our study were derived from this fully anonymized dataset.

We added the ethical statement in the method section 

We added these captions

Results

(lines 154-61): I find it quite interesting that the correlation is positively higher for asymptomatic cases than for symptomatic cases. Could the authors explain this finding?

We acknowledge this observation, and although we cannot offer a definitive explanation, we can put forth a plausible hypothesis. During the period from 2021-W33 to 2021-W40, we noted a significant decrease in the Positive Predictive Value (PPV) – and consequently, the estimated prevalence – among symptomatic individuals. Conversely, asymptomatic infections and hospitalizations experienced only a marginal decline. Moreover, a drop in PPV and estimated prevalence coincided with a sharp increase in hospital admissions from 2021-W42 to 2021-W49. We believe that a seasonal upswing in COVID-like symptoms caused by other pathogens may have contributed to this pattern. Additionally, during this period, the guidance to use self-tests exclusively when asymptomatic may have prompted individuals with multiple or severe symptoms to directly seek assistance from public health services, bypassing the use of Ag-RDTs. On the other hand, those with less distinctive symptoms (and consequently, a lower likelihood of SARS-CoV-2 infection) may have opted for self-testing instead. These factors are less relevant for asymptomatic individuals, potentially accounting for the stronger correlation observed in that group.

In the omicron period, we speculate that individuals with pronounced symptoms who received a positive Ag-RDT result may have refrained from seeking confirmation at public health services, further attenuating the correlation compared to asymptomatic individuals.

We added one sentence in our discussion to mention this fact. 

Under line (174-5) “An underestimation of the sensitivity and specificity of the Ag-RDT’s leads to an overestimation of the prevalence” : Is this sentence reporting the table results or is this a discussion point? If the former is true, I’m not sure if this is reporting the last 2 rows of Table 2? Also, I would suggest adding the sensitivity & specificity values for the baseline as a Table footnote.

This sentence refers to the table results. In the last two rows the table shows that increasing the true sensitivity and true leads to an overestimation of the prevalence using our method of analysis. We agree a footnote clarifies our statement and we added this footnote. 

Discussion

While the relationship between PPV and severity is emphasized in the introduction (and which is likely the basis why COVID-related hospital admissions data were collected), I feel there is a lack in discussing this point in the discussion section. One reason could be the structure of the section. This is especially true for the 6th paragraph, I’m quite confused why hospital admissions are being compared with prevalence. I would suggest to consider revising the discussion section to better clarify the points intended by the authors.

We agree with your statement. The central query we aimed to address in our paper is whether the Positive Predictive Value (PPV) of self-tests holds the potential to serve as a tool for estimating the underlying prevalence and so to contribute to discussions on disease severity. Our findings led us to conclude that while our current estimate proves valuable in discerning fundamental shifts in prevalence, it (currently) lacks the necessary precision for robust estimations of this prevalence to be used in context of hospitalization rates and infer disease severity. We concur that hospital admissions, being an incidence rate, are not a direct indicator of prevalence. Various other elements, such as age, immunity resulting from vaccination or prior infections, and evolving virulence across different variants, also exert influence on the number of hospital admissions. However we still think that hospital admissions is the best comparator available linked to the prevalence of infections and were only used to examine a correlation between our estimates of the prevalence based on PPV as an external validation. 

We changed our introduction and part of our discussion to better reflect the goals of our study and improved the overall structure. 

Suggest to include a brief explanation, or a timeline figure, on the public health intervention measures implemented in the Netherlands during the study period. As this would help readers understand better the local context and also the points raised in the discussion section.

We agree and we added a brief overview of measures in the Netherlands in the method section.

Minor comments:

- Abstract (2nd paragraph, 2nd sentence): Please clarify it’s the overall positivity rate of..?

We changed this to overall PPV. 

- The manuscript has some minor punctuation mistakes; I suggest to proofread carefully

We fixed some punctuation mistakes. 

- Table 1 caption: I don’t understand this, suggest to clarify: “***= All individuals with at least one vaccination that are not covered by the fully vaccinated category.”

These are individuals that only received one vaccination or received a second vaccination but were still within a 14-28 day window so they were not considered fully vaccinated. We changed the text in the table to clarify what we mean. 

- Table 2 (columns 5 & 6): Please check the headings if this should this be correctly read as “Propensity to test SARS-CoV-2 + / SARS-CoV-2 - ?” I would suggest to use positive & negative to avoid confusion, or add a footnote to clarify.

We agree and changed this in the text

Reviewer #1: Generally, well written manuscript. with the vailability of vaccination data for more than two thirds of study participants it would be great to get a sense of how vaccination coverage over time impacted estimated prevalence/ PPV in the vaccinated compared to the unvaccinated groups

We also found this an interesting question and in fact, we delved into this matter. During our research we concluded that without data on test propensity interpreting results of PPV in the context of prevalence is challenging, as a difference in PPV could occur due to differences in testing between vaccinated and unvaccinated individuals or due to a true difference in prevalence. Furthermore a difference in prevalence should depend on to which extent the vaccinated and unvaccinated socially mix as these individuals might cluster together and we don’t have data for this analysis. As we conclude in our paper PPV in the context of prevalence can be used to gauge underlying trends but is not yet suitable for precise interpretations of a subgroup analysis such as you suggest on vaccinated versus unvaccinated individuals as this necessitates additional data as mentioned above. 

For your interest we will give a rough overview of our analysis and results on vaccination status. 

Interestingly, the overall Positive Predictive Value and estimated prevalence (PPV) differed between vaccinated and unvaccinated individuals. From the period 2021-W27 (the first week data was available on vaccination status for vaccinated individuals) , unvaccinated individuals displayed a slightly higher PPV and estimated prevalence, but this trend reversed (approximately in 2021-W44) when the delta variant was dominant and subsequently remained when omicron became dominant. Given that we did not have data on test propensity we could not make a valid interpretation of this difference and why the difference reversed.

---

## [Decision Letter · Decision Letter 1]

22 Jan 2024

Estimating infection prevalence using the positive predictive value of self-administered rapid antigen diagnostic tests: an exploration of SARS-CoV-2 surveillance data in the Netherlands from May 2021 to April 2022

PONE-D-23-14636R1

Dear Dr. Gorgels,

We’re pleased to inform you that your manuscript has been judged scientifically suitable for publication and will be formally accepted for publication once it meets all outstanding technical requirements.

Kind regards,

Liling Chaw

Academic Editor

PLOS ONE

Additional Editor Comments (optional):

Reviewers' comments:

Reviewer's Responses to Questions

**Comments to the Author**

1. If the authors have adequately addressed your comments raised in a previous round of review and you feel that this manuscript is now acceptable for publication, you may indicate that here to bypass the “Comments to the Author” section, enter your conflict of interest statement in the “Confidential to Editor” section, and submit your "Accept" recommendation.

Reviewer #2: All comments have been addressed

2. Is the manuscript technically sound, and do the data support the conclusions?

Reviewer #2: Yes

3. Has the statistical analysis been performed appropriately and rigorously? 

Reviewer #2: Yes

4. Have the authors made all data underlying the findings in their manuscript fully available?

Reviewer #2: Yes

5. Is the manuscript presented in an intelligible fashion and written in standard English?

Reviewer #2: Yes

6. Review Comments to the Author

Reviewer #2: I have no further comments and after extensive review the authors adequately addressed all issues with the manuscript.

7. PLOS authors have the option to publish the peer review history of their article (what does this mean?). If published, this will include your full peer review and any attached files.

Reviewer #2: No

---

## [Editor Report · Acceptance letter]

23 Jan 2024

PONE-D-23-14636R1 

PLOS ONE

Dear Dr. Gorgels, 

I'm pleased to inform you that your manuscript has been deemed suitable for publication in PLOS ONE. Congratulations! Your manuscript is now being handed over to our production team.

Kind regards, 

on behalf of

Dr. Liling Chaw 

Academic Editor

PLOS ONE